# A Pilot Study to Evaluate the Usefulness of Optical Coherence Tomography for Staging Iris Pigmented Lesions in Cats

**DOI:** 10.3390/vetsci11060261

**Published:** 2024-06-07

**Authors:** Hiroyuki Komatsu, Minori Akasaka, Maresuke Morita, Kensuke Usami, Mao Inagaki, Kayo Kumashiro, Kinya Tsubota, Yoshihiko Usui, Hiroshi Goto, Yoshitaka Kobayashi

**Affiliations:** 1Animal Eye Care Tokyo Animal Eye Clinic, 1-1-3 Kaminoge, Setagaya, Tokyo 158-0093, Japancoba@animaleyecare.jp (Y.K.); 2Department of Ophthalmology, Tokyo Medical University, 6-7-1 Nishishinjuku, Shinjuku, Tokyo 160-0023, Japan

**Keywords:** feline diffuse iris melanoma, iris melanosis, ocular imaging, optical coherence tomography

## Abstract

**Simple Summary:**

Feline diffuse iris melanoma (FDIM) is the most common uveal tumor that develops within cats worldwide; thus, surgical resection is recommended at an early stage of the disease. However, it is difficult to differentiate early-stage FDIM from iris melanosis, a precursor lesion, based on clinical findings alone. In this study, we investigated the effectiveness of optical coherence tomography (OCT) in differentiating early FDIM from iris melanosis. The results of this study provide useful information for clinicians struggling with the timing of surgery. OCT has the potential to be an adjunctive diagnostic tool to differentiate iris melanosis from early-stage FDIM.

**Abstract:**

This study investigated the utility of optical coherence tomography (OCT) for staging iris pigmented lesions in cats. Eighteen cats that underwent OCT examination for unilateral iris pigmented lesion were included. The cats were either suspected of melanosis due to clinical features (*n* = 8) or had been definitively diagnosed through histopathology with iris melanosis (*n* = 3), early feline diffuse iris melanoma (FDIM) (*n* = 4), or mid-stage or advanced FDIM (*n* = 3). From OCT images, mean iris thickness (MIT) was measured, and the ratio of pigmented lesion to normal iris (PN) was calculated. OCT images depicted the entire iris layer in all eyes with suspected melanosis, iris melanosis, and early FDIM, but observing the entire lesion in mid-stage/advanced FDIM was challenging. No significant difference in MIT was observed among the groups. Conversely, PN ratio was significantly higher (*p* < 0.05) in early FDIM (1.29 ± 0.16) than in suspected melanosis (1.02 ± 0.10) or iris melanosis (0.99 ± 0.09). Furthermore, OCT imaging revealed hyperreflective lines in 75% of eyes with suspected melanosis and in all the eyes with iris melanosis, corresponding to the pigmented lesions. Our results demonstrate that OCT is capable of detecting subtle differences in iris thickness and features in early-stage FDIM, indicating its potential utility in distinguishing between iris melanosis and early FDIM. Further study is warranted to verify the reliability of such OCT findings.

## 1. Introduction

Feline diffuse iris melanoma (FDIM) represents the most prevalent form of intraocular malignancy in cats [1,2,3]. The pathophysiology of FDIM is complex. FDIM in the early stage rarely metastasizes, whereas advanced FDIM has been documented to metastasize to systemic organs including the liver, lungs, spleen, lymph nodes, and bones in 19% [4] and 66% [5] of cats studied. A study of survival in cats with enucleation due to FDIM showed that while cats with early disease at enucleation survived comparably to controls, cats with advanced disease at the time of enucleation had significantly decreased survival intervals [5]. Currently, no effective chemotherapy or radiotherapy regimens exist, making surgical resections including enucleation and local excision the sole first-line treatment option [2]. Regarding the pathophysiological context of FDIM, iris melanosis, which is recognized as the precursor stage, is characterized by cell proliferation on the iris surface. These cells, which are darkly pigmented with uniform morphology and small, round nucleoli, accumulate on the iris. FDIM develops from these benign lesions through a process of malignant transformation over several months to years [3]. This transformation, marked by cells with diverse morphologies including angular shapes and dysplasia, leads to subsequent invasion into the iris stroma. As the disease progresses, the invasion extends to involve the entire iris, eventually reaching the ciliary body, followed by the iridocorneal angle and sclera. At this stage, clinical findings such as an irregular iris surface and secondary glaucoma may be observed [6]. Thus, histopathological findings allow distinct differentiation of the transition from melanosis to early FDIM, based on the degree of invasion and cellular morphology. However, clinical examinations alone pose significant challenges in differentiating between melanosis and early-stage FDIM, as both conditions manifest focal or multifocal flat, hyperpigmented areas on the iris. While melanosis can be managed with observation alone due to its benign nature, early enucleation (or iridectomy) has been recommended to be important in cats with FDIM to avoid premature death due to cancer metastasis. However, histopathological evaluation is necessary to differentiate between melanosis and early FDIM, and this requires a surgical procedure. In addition, owners are sometimes reluctant to consent to enucleation of an eye with residual vision, especially when the contralateral eye has lost vision. Consequently, due to this background, many veterinary surgeons have difficulties deciding the timing of surgical intervention. Such factors may delay timely treatment intervention, potentially resulting in increased metastatic risks and shortened survival. Determining malignant transformation from melanosis to FDIM based solely on clinical findings is frequently difficult, which is a significant obstacle for determining the timing of surgical intervention. Accurately establishing the optimal timing for treatment of early-stage FDIM remains a pivotal issue in clinical practice.

In recent decades, optical coherence tomography (OCT) has been developed in the field of ophthalmology, and the usefulness of OCT for the evaluation of the anterior or posterior segment has been widely reported in human medicine [7,8,9]. OCT is a useful device capable of providing high-resolution and cross-sectional images of ocular tissues, thereby enabling detailed observation of tissue structures and noninvasively obtaining findings consistent with histopathological characteristics. In human ophthalmology, there are various reports of anterior segment OCT (AS-OCT) for the evaluation of iris lesions including tumors [10,11,12,13]. These reports suggest that AS-OCT has the advantage of being a less complicated non-contact clinical examination requiring minimal technical skills, although the image quality is generally considered inferior to that of ultrasound biomicroscopy (UBM) [14]. A report comparing AS-OCT with UBM showed that while UBM offered better visualization of the posterior margin, AS-OCT provided better resolution of the anterior margin of the tumor than UBM [13]. In veterinary medicine, an increasing number of reports on the use of OCT for evaluating the anterior segment including the cornea, anterior chamber, and iridocorneal angle in dogs have been published [15,16,17,18]. However, there are no clinical reports focusing on the use of AS-OCT for examining iris lesions in cats.

This background prompted us to conduct a pilot study to investigate whether OCT can be used to differentiate FDIM from iris melanosis in cats. If OCT proves to be effective for early diagnosis of FDIM, the findings may resolve the challenges of distinguishing melanosis from FDIM, providing a new approach for determining the optimal timing for treatment of FDIM in veterinary practice.

## 2. Materials and Methods

### 2.1. Cases

This study enrolled 18 cats in which AS-OCT examinations were performed for investigation of unilateral iris hyper-pigmented lesions at Animal Eye Care Tokyo Animal Eye Clinic between 2018 and 2023. The cats were either suspected of melanosis based on clinical features (8 cats) or were definitively diagnosed via histopathology as having iris melanosis (3 cats), early stage FDIM (4 cats), or mid-stage or advanced-stage FDIM (3 cats). Histopathological examination was conducted on the surgical tissues from all cats that underwent enucleation or iridectomy. FDIM was diagnosed and staged by a pathologist in accordance with the histopathological criteria described in the literature [19]. Briefly, iris melanosis is characterized by localized pigmented lesions originating from clusters of small, angular cells with round nuclei on the iridal surface. Early-stage FDIM features tumor cells with large, round nuclei and prominent nucleoli invading the iridal stroma. As the disease progresses, the lesion extends to the ciliary body in the mid-stage and invades the sclera and extraocular tissues in the advanced stage. All the cats underwent complete ophthalmologic examination including slit-lamp biomicroscopy (BQ-900 LED, Haag-Streit AG, Köniz, Schweiz), Schirmer tear test 1, rebound tonometry (TONOVET or TONOVET Plus, Icare, Vantaa, Finland), fluorescein staining, indirect ophthalmoscopy (Omega 500, HEINE, Gilching, Germany), and ultrasonography (ARIETTA 60 device, Aloka Corporation, Tokyo, Japan). Ophthalmologic examinations were performed by diplomates of the Japanese Society of Comparative and Veterinary Ophthalmology (JSCVO) and by veterinary staff trained by JSCVO diplomates. In addition, all the cats underwent a comprehensive systemic examination including thoracic radiography or abdominal ultrasonography, which revealed no apparent neoplastic lesions in other organs. Furthermore, apart from the pigmented lesions, no other ophthalmic diseases such as uveitis and glaucoma were detected, based on the absence of anterior chamber flare and the intraocular pressure. Detailed data of the cats in the five groups are presented in Table 1. This study was reviewed and approval by the institutional review board of Tokyo Animal Eye Clinic (No. T2023-001). Informed consent was obtained from the owners of all the cats studied, and the study was conducted in compliance with the Association for Research in Vision and Ophthalmology (ARVO) Statement on the Use of Animals in Ophthalmology and Vision Research.

### 2.2. Measurements of Iris Thickness Using OCT

OCT images of all the eyes were acquired using spectral domain OCT (Spectralis, Heidelberg) with the anterior segment module [20] (Figure 1). For the purpose of evaluating iris lesions under conditions of pupil contraction, OCT examinations were conducted under lighting with an illuminator (SL-17, KOWA, Aichi, Japan) directed at the contralateral eye. OCT examinations were conducted by diplomates of JSCVO and veterinary staff trained by diplomates of JSCVO. The head of the cat was manually restrained in front of the OCT device, without using sedation or anesthesia. From the OCT images obtained, the thickness of the iris in the pigmented lesion was measured at three locations: the iris root, iris margin, and center of the pigmented lesion, and the mean iris thickness (MIT) of the pigmented lesion was calculated. In addition, the MIT of the pigmented lesion in the affected eye and the MIT of the corresponding area in the unaffected contralateral eye were measured, and the ratio of the pigmented lesion to the normal area (PN ratio) was calculated (details described in Appendix A). In cases where iris melanosis was observed at multiple sites, several measurement locations were randomly selected from areas that permitted comparison with OCT images of the contralateral eye. Thickness measurements were conducted on OCT images extracted on a 1:1 micrometer basis using ImageJ software ver. 1.54f, after calibrating the distance in pixels to the 200 μm scale bar displayed on the images. MIT and PN ratio were calculated by one observer (H.K.). OCT examination was performed within 1 month before surgery in all cats that underwent enucleation or local excision.

### 2.3. Statistical Analysis

Statistical analyses were conducted using GraphPad Prism 10 (version 10.1.1) and SPSS (IBM, Armonk, NY, USA). Data are expressed as mean ± standard deviation. Comparisons of iris thickness and PN ratio among the groups were performed using one-way ANOVA followed by Tukey’s honestly significant difference test to adjust for multiple comparison. Normality of the data was assessed using the Shapiro–Wilk test. Adjusted *p*-values less than 0.05 were considered statistically significant. Intraclass correlation coefficients were calculated to assess reproducibility, with intraobserver reliability determined through performing three repeated measurements of each sample including pigment and normal lesion (*n* = 30) taken by one observer (H.K.) and interobserver reliability assessed through performing single measurements of each sample including pigment and normal lesion (*n* = 30), carried out by two observers (H.K. and M.M.). The 95% confidence interval (CI) of the *p*-value was calculated.

## 3. Results

The clinical data of the cats with definitive diagnoses confirmed via histopathology are listed in Table 2. Cats with suspected melanosis showed no significant increase in lesion size during a median follow-up period of 591 (range: 0–2104) days. On the other hand, surgical excision was performed in cats with iris melanosis or early FDIM due to slight enlargement of the lesion during follow-up or the owner’s choice. Slit-lamp examination revealed no obvious iris thickening or irregular iris surface in the pigmented lesions in cats with suspected melanosis or those with iris melanosis. On the other hand, among cats with early FDIM, Cases 5 and 6 showed flat and smooth pigmented lesions, Case 4 exhibited slight focal bulging of the iris in the pigmented lesion, and Case 7 showed increased iris thickness and irregular iris surface in the pigmented lesions. Ultrasonography detected no noticeable iris thickening or suspected iris findings in any of the cats.

OCT examination of the iris pigmented lesion was completed within 10 min in all the eyes. In the eyes with suspected melanosis, iris melanosis, or early FDIM, the acquired OCT images successfully depicted the iris surface and the iris epithelial layer, facilitating the evaluation of MIT. However, in the eyes with mid-stage or advanced FDIM, delineating the iris pigment epithelial surface was difficult due to the severe shadow effect, hindering the evaluation of the entire iris (Figure 2).

The location of the pigmented lesion, iris thicknesses measured at the iris root, center, and margin of the pigmented lesion, calculated MIT in the pigmented lesion, calculated MIT in the contralateral eye, and calculated PN ratio for each cat are shown in Appendix A.

MIT was compared among the suspected melanosis, iris melanosis, and early FDIM groups in which MIT was measurable. The group average MIT of the lesions was 290 μm (mean ± standard deviation) in the suspected melanosis group, 352 ± 95 μm in the iris melanosis group, and 353 ± 105 μm in the early-FDIM group (Figure 3). No significant difference in the MIT of lesions was observed among these three groups (*p* = 0.61, one-way ANOVA). 

The PN ratio was also compared among the three groups. The PN ratio was 1.02 ± 0.10 in the suspected melanosis group and 0.99 ± 0.09 in the iris melanosis group, indicating that the thickness of the pigmented lesions was almost the same as the normal tissue in the contralateral eye in both groups. In contrast, the early FDIM group exhibited a PN ratio of 1.28 ± 0.18, demonstrating a slight increase in thickness in the pigmented lesion compared with that in normal iris in the contralateral eye. Especially, a significant difference in PN ratio among the three groups was detected (*p* < 0.05, one-way ANOVA). Furthermore, Student’s t-test detected a significantly higher PN ratio in the early FDIM group compared with both the suspected melanosis group (adjusted *p* < 0.05, 95% CI: 0.059 to 0.491) and the iris melanosis group (adjusted *p* < 0.05, 95% CI: 0.037 to 0.576) (Figure 3). 

Intraclass correlation coefficients demonstrated high reliability for both intraobserver and interobserver reproducibility for MIT (0.996 and 0.839, respectively) and PN ratio (0.988 and 0.896, respectively) (Appendix A).

In some eyes, the OCT images showed hyperreflective lines with higher intensity than the iris stroma in the area corresponding to the pigmented lesion (Figure 4). These hyperreflective lines were observed in 6 cats (75%) in the suspected melanosis group and 3 cats (100%) in the iris melanosis group, but in none of the cats in the early FDIM or the mid-stage/advanced FDIM groups. Histopathological examination of the pigmented lesions in the eyes with iris melanosis revealed an accumulation of pigmented cells in the area corresponding to the location of the hyperreflective line, whereas no accumulation of pigmented cells on the iris surface was observed in all the eyes with early FDIM.

## 4. Discussion

In this pilot study investigating the usefulness of OCT for evaluating iris pigmented lesions in cats, full-thickness images of portions of pigmented areas were obtained from eyes with iris melanosis and eyes with early FDIM. Our study found a significant relative increase in iris thickness at the site of the pigmented lesion in early-FDIM eyes compared with iris melanosis eyes. Furthermore, the hyperreflective line delineated via OCT was only detected in eyes with iris melanosis and not in early-FDIM eyes, indicating that OCT may reflect a specific feature of melanosis, corresponding to melanocyte accumulation as revealed by histopathological examination. This is the first report using OCT to evaluate iris pigmented lesions in cats.

In human ophthalmology, the advent of OCT has brought a paradigm shift in diagnosis and development of treatment strategies due to its ability to depict high-resolution images of tissues [21,22]. In particular, OCT has become an indispensable tool in the management of glaucoma [23,24,25] and various retinal diseases [26,27,28,29,30], contributing to the understanding of pathophysiology and the establishment of parameters related to vision prognosis, thereby informing the determination of treatment strategy [26]. AS-OCT is also commonly used to evaluate corneal shape and iridocorneal angles [31,32]. Spectral domain OCT, which is one of the methods of OCT, operates at a wavelength range of about 820–840 nm [8,31]. In our current study using SD-OCT, while evaluation of the full thickness of the iris in feline eyes with iris melanosis and early FDIM was feasible, assessing the full layers of the iris in mid-stage and advanced FDIM proved challenging. However, since iris thickening and irregularity are assumed to have progressed in mid-stage FDIM and advanced FDIM, these abnormalities may be observable in slit-lamp examination and ultrasonography. Hence, SD-OCT has the potential to evaluate the subtle difference in iris thickness between iris melanosis and early FDIM, as demonstrated in this study. Moreover, SD-OCT has the significant advantage of faster image acquisition, which is crucial since animals often have difficulties maintaining eye fixation during OCT examination without anesthesia. Therefore, SD-OCT may be more suitable for evaluating iris pigmented lesions in cats in the clinical setting.

Clinical examination for FDIM is currently performed using slit-lamp or ultrasonography [2]. If the slit-lamp examination or ultrasonography reveals noticeable iris thickening or an irregular iris surface in the pigmented lesion, tumorigenesis at the site of iris pigmentation is highly suspected. However, differentiating between iris melanosis and early FDIM using slit-lamp or ultrasonography is challenging, as mentioned above, as both conditions present as flat pigmented areas [3]. In the current study, OCT detected an approximately 1.3-fold increase in iris thickness in early FDIM lesions compared with normal irises. Considering that the difference in iris thickness between early FDIM and melanosis is only a few tens of micrometers, as observed in this study, this finding explains the fact that such a difference is undetectable via slit-lamp or ultrasonography because the resolution of ultrasonography is approximately 100 nm. Recently, iris biopsy has been reported to be useful for the diagnosis of melanosis before surgery [33]. Iris biopsy is useful for localized lesions because it allows a definitive histopathological diagnosis, which avoids unnecessary enucleation. In eyes with multiple pigmented lesions, however, it is difficult to perform biopsies on all the affected sites. Furthermore, periodic biopsies are generally needed to monitor tumor progression over time, a process that may be highly invasive. In contrast, OCT can be performed quickly and noninvasively, allowing evaluation of multiple sites over time, even in eyes with multiple lesions. Thus, OCT may serve as an adjunctive tool to monitor changes in iris structures over time in eyes with multiple iris pigmented lesions.

In this study, there was no significant difference in the MIT of the lesions among the groups. Conversely, there was a significant increase in the PN ratio in early FDIM compared with suspected or confirmed iris melanosis. This discrepancy may be attributed to the characteristic anatomy of the iris, which varies in thickness from the iris root to iris margin [34]. As a result, MIT measurement of lesions is greatly influenced by the location of the lesions on the iris, potentially masking subtle changes in iris thickness in early FDIM. In the current study, subtle changes in early FDIM were detected using the ratio of the iris thickness in the pigmented lesion to that in normal iris tissue in the corresponding area of the contralateral eye. Evaluation using methods comparing lesions to normal areas, such as the PN ratio in this study, may be effective for assessing iris pigmented lesions in OCT examinations. 

Interestingly, this study observed hyperreflective lines on OCT images corresponding to the pigmented lesions in eyes with iris melanosis. This finding suggests an association with the specificity of OCT, which operates on the principle of interferometry, directing a light beam at target tissue and analyzing the interference pattern between reflected tissue light and a reference beam [7]. Hence, OCT has the advantage of the ability to reflect variations in tissue cell density and composition. In cats with pigmented lesions, pathophysiological differences exist between iris melanosis and early FDIM. Pathological findings of iris melanosis include accumulation of pigmented cells on the iris surface without invasion into the iris stroma, which is mainly composed of connective tissue with low cell density. Conversely, early FDIM is characterized by tumor cell infiltration and proliferation in the iris stroma, resulting in less pronounced contrast in cell density between the iris surface and stroma due to tumor infiltration [35]. Therefore, the contrast between light-impermeable pigment cells and low-density iris stroma is more pronounced in iris melanosis, possibly resulting in formation of the observed hyperreflective lines. The OCT findings were consistent with the histopathological observations in this study, suggesting that OCT can potentially detect the pathological characteristics of iris melanosis as hyperreflective lines.

This study had several limitations, with the small sample size being the most significant issue. Although FDIM is the most common intraocular tumor in cats, this condition is frequently referred to specialized ophthalmological facilities at advanced stages. One reason is that both owners and primary care veterinarians are unaware of melanosis or early FDIM lesions, because of the lack of symptoms. This limits the opportunity to evaluate melanosis and early-stage FDIM using OCT. Furthermore, obtaining a definitive histopathological diagnosis of melanosis after enucleation or local excision is exceedingly rare during the melanosis stage, posing a significant challenge in expanding the sample size. Additionally, as this pilot study was conducted at a single center, it did not account for regional and breed differences in iris morphology, which limits the generalizability of the findings. Moreover, this study did not evaluate all multiple lesion sites or longitudinal changes. In addition, given the difficulty in assessing past history of uveitis, the possibility of residual effects from uveitis cannot be ruled out. Furthermore, the resolution of the OCT images was compromised in some cases in this study, because sedation or anesthesia was not used, thus limiting the number of frames of OCT images that could be captured. When more detailed evaluation is necessary, using anesthesia or sedation to immobilize the eye may enhance the image quality by increasing the number of image frames and allow observation of multiple lesion sites. Given the limitations of this study, OCT alone did not differentiate between melanoma and melanosis. Staging decisions should be made carefully and comprehensively, based on clinical findings, the disease course, and auxiliary tools such as OCT images. Moreover, given that the results of this study may not be generalizable to all cases, further research involving a larger cohort is indispensable. Future research, ideally in the form of international multicenter studies, is needed to enroll a larger sample for more precise evaluations. Regarding methodology, our findings indicate that mid- or advanced-stage FDIM cannot be evaluated by OCT examination. Based on our findings, future research with larger sample sizes and rigorous measurement and statistical methods is essential to ensure the robustness of the findings. 

## 5. Conclusions

In conclusion, this study in cats shows that OCT is capable of capturing subtle differences in iris thickness in early-stage FDIM compared with confirmed iris melanosis. Cats with suspected melanosis showed similar OCT findings as cats with confirmed melanosis in this study. Hyperreflective lines observed on the iris surface were likely to capture histopathological features in eyes with iris melanosis. Therefore, the results indicate that OCT has the potential to be a new auxiliary tool to stage iris pigmented lesions in cats and differentiate early FDIM from iris melanosis, which has been difficult in clinical examinations using slit-lamp and ultrasonography. However, it is difficult to determine the exact nature of iris pigmentation based on AS-OCT alone. Further in-depth studies with more cats, correlating findings of OCT with histopathology and other image modalities, are needed to better understand iris pigmented lesions in cats.

Thus, the present results support the possibility that OCT may become a new adjunctive tool in disease classification of iris pigmented lesions, previously carried out via slit examinations and ultrasound examinations.

## Figures and Tables

**Figure 1 vetsci-11-00261-f001:**
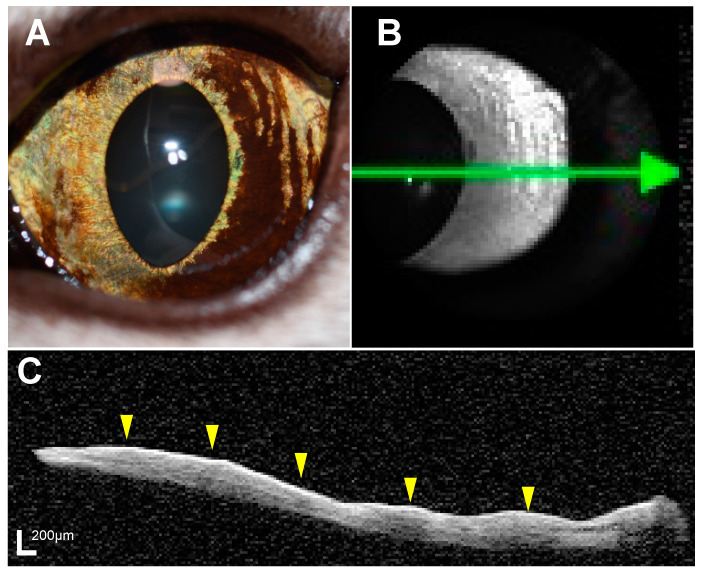
Representative anterior segment photograph and optical coherence tomography (OCT) images. A cat that was histopathologically diagnosed with iris melanosis showed multiple diffuse, well-demarcated pigmented lesions (**A**). On OCT examination, infrared imaging (**B**) showed an indistinct boundary between the pigmented and non-pigmented areas. However, the OCT image (**C**) demonstrated a hyperreflective line on the iris surface and a mild shadow effect in the iris stroma corresponding to the pigmented lesions (yellow arrowheads). The green line on the infrared image represents the area captured in the cross-sectional view of the OCT images, with the green arrowhead pointing to the right side on the OCT image.

**Figure 2 vetsci-11-00261-f002:**
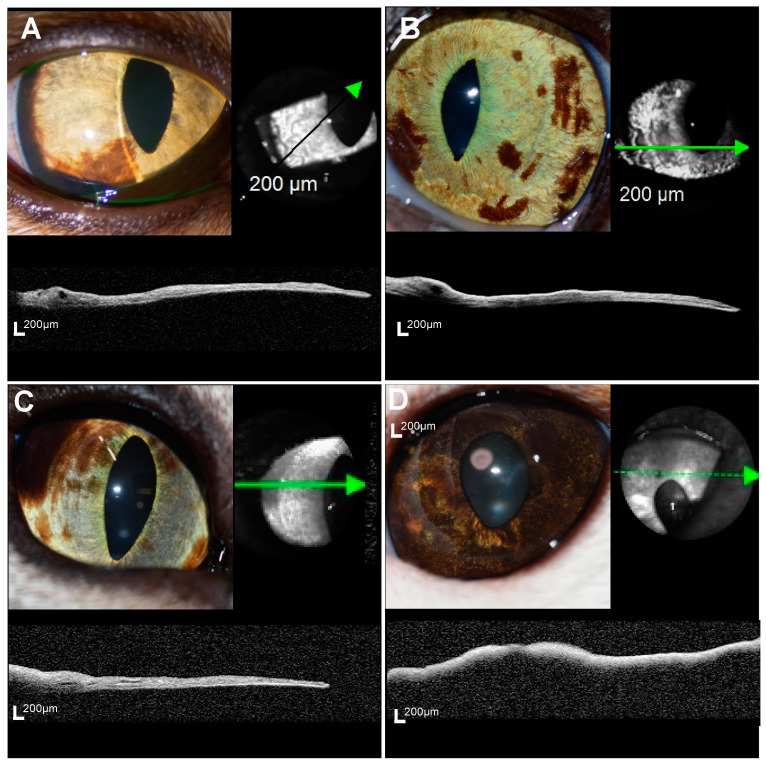
Representative anterior segment photographs and OCT images for four groups. In the cats with suspected melanosis (**A**), iris melanosis (Case 1) (**B**), and early FDIM (Case 4) (**C**), the iris pigment epithelium could be observed and total iris thickness was evaluated. However, in the cat with advanced stage FDIM (Case 9) (**D**), complete evaluation of the total iris layer was not possible due to severe shadowing artifacts. The green line, black line, and green dot line on the infrared image represent the area captured in the cross-sectional view of the OCT images, with the green arrowhead pointing to the right side on the OCT image.

**Figure 3 vetsci-11-00261-f003:**
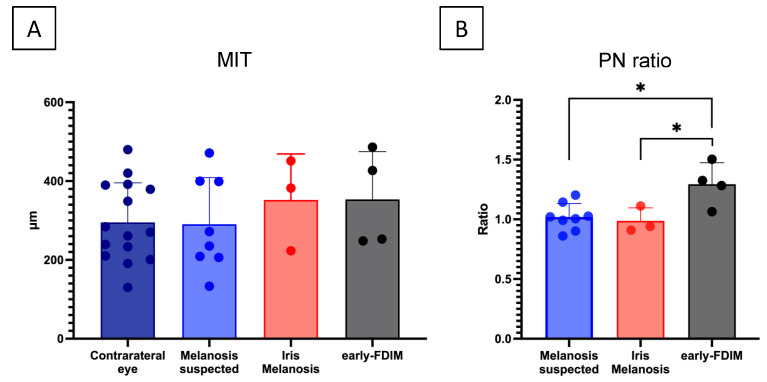
Comparison of mean iris thickness of total layer of pigmented lesion and ratio of thickness of pigmented lesion to thickness of normal iris (PN) in the three groups (**A**). No significant difference in the mean iris thickness of the full layer was observed among the three groups (**A**). Conversely, the (PN) ratio was significantly higher in the early FDIM group compared with the suspected melanosis group and iris melanosis group (**B**). * *p* < 0.05, Tukey’s honestly significant difference test.

**Figure 4 vetsci-11-00261-f004:**
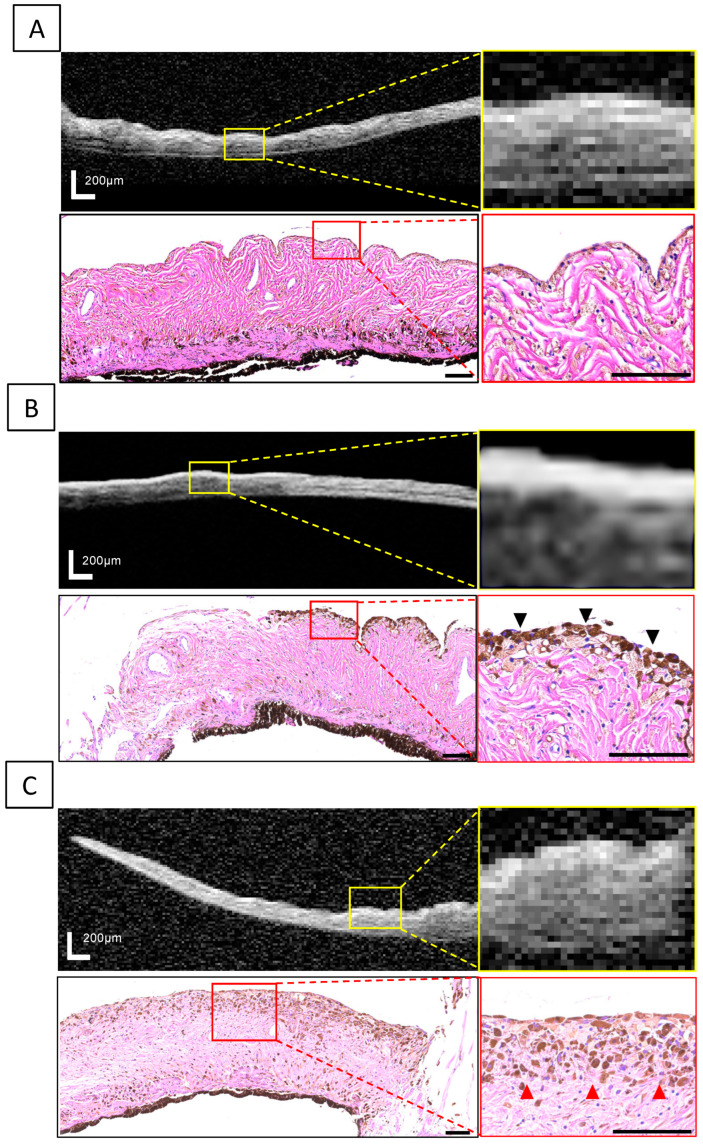
Comparison of OCT images and histopathological findings in a non-affected iris (**A**), iris melanosis (**B**), and early FDIM (**C**). In the cat with iris melanosis, a hyperreflective line corresponding to the pigmentation area was delineated on the OCT image, and histopathological examination showed an accumulation of pigmented cells with uniform morphology and scarce dysplasia on the iris surface (black arrows). On the other hand, in the cat with early FDIM, no hyperreflective line was found in the pigmented lesions, and histopathological findings confirmed infiltration of tumor cells with dysplasia into the iris stroma (red arrows). Scale bar: 200 μm in OCT images and 100 μm in histopathological images.

**Table 1 vetsci-11-00261-t001:** Background data of the cats in each group in this study.

		Suspected Melanosis	Iris Melanosis	Early FDIM	Mid-Stage FDIM	Advanced FDIM
Number of cats		8	3	4	1	2
Age (Year, Mean ± SD)		6.3 ± 3.2	6.0 ± 3.8	3.6 ± 2.1	11	11.8 ± 1.3
Breed						
	Mixed	5	1	2	1	2
	Scottish Fold	0	1	1	0	0
	Exotic Shorthair	1	1	0	0	0
	American Shorthair	0	0	1	0	0
	Munchkin	1	0	0	0	0
	Russian Blue	1	0	0	0	0
Sex						
	Neutered male–spayed female	4:4	3:0	2:1	0:1	1:1
Median follow-up period						
	(Days [range])	591 (0–2104)	29 (28–478)	89 (9–202)	118	29 (27–30)

SD; standard deviation, FDIM; feline diffuse iris melanoma.

**Table 2 vetsci-11-00261-t002:** Clinical data of the cats definitively diagnosed with iris melanosis or FDIM confirmed via histopathology.

Case No.	Diagnosis	Laterality	Location	Surgery
1	Iris melanosis	Left	Multi-focal	Enucleation
2	Iris melanosis	Right	Multi-focal	Enucleation
3	Iris melanosis	Left	Multi-focal	Enucleation
4	Early FDIM	Left	Focal in iris root	Enucleation
5	Early FDIM	Right	Focal in iris root	Local resection
6	Early FDIM	Right	Multi-focal	Enucleation
7	Early FDIM	Left	Entire iris	Enucleation
8	Mid-stage FDIM	Left	Entire iris	Enucleation
9	Late-stage FDIM	Left	Entire iris	Enucleation
10	Late-stage FDIM	Right	Entire iris	Enucleation

FDIM; feline diffuse iris melanoma.

## Data Availability

The data presented in this study are contained within the article.

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
