# Peer review of "A Pilot Study to Evaluate the Usefulness of Optical Coherence Tomography for Staging Iris Pigmented Lesions in Cats"

_vetsci, 2024, doi:10.3390/vetsci11060261_

Round 1

Reviewer 1 Report

Comments and Suggestions for Authors

These authors identify a pivotal dilemma in feline ophthalmology.  These authors do capture a pivotal dilemma in feline ophthalmology.  Moreover, this reviewer applauds and supports all scientifically rigorous efforts to better understand the clinical approach to this disease.  However, the reason for the relatively small body of literature regarding FDIM is largely because sound study design for this disease is complicated; and methods that lack scientific rigor, like those presented in this manuscript, do not meaningfully add to that body of literature. This reviewer is particularly concerned that the data presented are derived from very flawed methodology that is susceptible to bias and poor reproducibility.  The conclusions offered by these authors over-state the results; and the statement that “OCT captures subtle changes in iris thickness in early-stage FDIM compared with iris melanosis and suspected melanosis” is not supported by the data.  As written in their current form, these conclusions imply that the described OCT approach provides an easy solution to a highly complex clinical dilemma which is not supported by the data. This author is also very concerned that these authors included cases of “suspected melanosis” in tandem with those confirmed by histopathology – this is scientifically untenable and reduces the validity of the data and conclusions. The authors’ background information also includes inaccuracies and misleading statements regarding the standards of care for and fundamental knowledge regarding FDIM. This reviewer encourages these authors to pursue more rigorous methods moving forward, inclusion of more patients, and use of cases with verified histologic diagnosis of FDIM. Please also consider a more objective evaluation of inter- and intraobserver reproducibility of your OCT methods and measurements. Please see more specific comments below.

Background Information/Introduction Notes

Stating that FDIM has a poor prognosis and “that surgical resection is recommended at early stage of the disease” is not in line with the weight of the evidence in the literature. Can the authors provide a reference that specifically states this? This clinical approach may be the standard at the authors’ practice, but is not necessarily shared as a universal approach elsewhere.  

The authors make the blanket statement in several locations that FDIM has a “poor prognosis”.  However, this is an oversimplification of a very complex pathophysiology that is not so “black and white”.  Moreover, references #1-3 that the authors cite as support for this statement do not specifically make the same statement.  The second sentence in the paragraph more accurately presents the more widely-accepted suggestion that advanced FDIM may be associated with higher risks of metastasis. There are conflicting statements even within the same paragraph that are confusing to the reader. The authors indicate in the third sentence that FDIM in the early stage is rarely associated with metastasis and that this leads to a “good prognosis”.  This is confusing and potentially misleading information to practitioners.

The authors include a great deal of information about FDIM pathophysiology and pathologic features.  Much of the detailed information about cellular features is unnecessary as the OCT imaging data presented later in the manuscript only examines thickness measurements, thickness ratios, and descriptive data regarding “hyperreflective lines”. 

The authors’ use of the term “precancerous” for iris lesions is flawed as not all lesions go on to become neoplasia.  From reference #3: “The behaviour of these lesions is unpredictable. Some remain static or grow slowly over months to years, resulting in only a cosmetic change to the iris”.  The term ““precursor” lesions as used in reference #3 is more appropriate.

The authors indicate that progression of FDIM involves eventual involvement of the ICA and sclera.  While these structures may become involved with progression, one of the more reliable and established histologic criteria is progression to involve the ciliary body which the authors do not address at any point in the manuscript.

“Iris bulging” is not a universally observed finding in all advanced cases of FDIM. Can the authors provide a reference for this?

The references chosen to illustrate OCT development and use for anterior segment evaluation in human patients don’t seem to be the most appropriate; and 2 of those 3 are very outdated. The authors should try to find more timely articles.

The authors use references 9-12 to support the use of AS-OCT for evaluation of iris tumors in human patients.  However, 3 of these publication specifically identify the inferiority (or at least no overt advantage) of AS-OCT in comparison to UBM for evaluating iris neoplasms.  This reviewer acknowledges that the authors are more focused on the anterior iris margin which is supported by reference 13.  However, the authors should be more cautious about reading the details of the references they cite as they may actually contradict some of their conclusions, particularly regarding value of UBM compared to AS-OCT.  This can also be confusing to the reader.

Materials & Methods

The authors present their data for 18 cats. However, this “n” is misleading.  Three cats had mid- or advanced-stage FDIM and their measurements were not really even included.  So really, 15 cats were evaluated.  One of this reviewer’s primary concerns is that n=8 cats did not have histologically-confirmed melanosis.  So, over half of the lesions measured in this study (8/15) did not have verified melanosis, leaving only 7 where a diagnosis was confirmed.  This is a major design flaw that alone disqualifies many of the authors’ conclusions.

The authors indicate use of pathologic staging criteria according to reference 18.  However, this chapter only includes detail about imaging and not pathologic staging criteria.  The authors need to address this discrepancy.

What is a “comprehensive systemic examination”. Does this include thoracic radiographs and/or abdominal ultrasound?  The authors need to be more specific.

How were uveitis and/or glaucoma ruled out? This question may seem pedantic but these details would be important to the readership of this journal.

Why are the authors including information about median follow-up time if this was not a parameter of interest in the rest of the study.  There is no other mention of monitoring of any patient in this manuscript so this information does not related to anything else and is confusing.

Measurements of Iris Thickness

This reviewer has significant concerns about the methods undertaken for iris thickness measurement.  The authors indicate that they selected three locations within pigmented lesions.  However, the authors specifically state in their methods and show in figures (i.e. Figure 1) that some cats had multiple lesions.  In those cases, how did the authors determine the “center” of a pigmented lesions.  There is really no way to normalize these measurements across patients due to the differences in lesion distribution.

This reviewer is also highly concerned that the methods in the primary endpoints are subject to bias and poor reproducibility.  A better “pilot study” would be aimed at assessing inter- and intra-observer reproducibility of these methods rather than jumping right to making very large clinical conclusions.  How do the authors justify reproducibility between the MIT of a pigmented lesion and the corresponding location in the contralateral iris.  Also, did the authors use the same light stimulation in the eye with the lesion when evaluating the normal contralateral eye? This is not specifically stated.

The authors later identify differences in PN ratio as the pivotal finding supporting use of AS-OCT to differentiate early FDIM from iris melanosis.  Despite the statistical significance, the difference was exceptionally small and could very likely be influenced by the methods and how they were carried out, particularly with such a small sample size. 

Lack of consistency could have a VERY CONSIDERABLE effect on your data, particularly PN ratio, considering the relatively small difference between early FDIM and confirmed melanosis and also the very small sample size.

It is unclear how the authors actually used ImageJ in their methods.  They mention use of pixels for facilitating thickness measurements; but how?  More information is required.

Statistical Analysis

Statistically, was mean chosen because the data was normally distributed? Normally, such a low sample size might be better communicated using median and range or interquartile range.

Were multiple comparisons analyses (i.e. Tukey’s?) incorporated for the ANOVA and T-tests? The authors compared several groups so this would likely be a more appropriate statistical approach.

Results

Figure 1 C does not clearly illustrate the “hyperreflective lines” that the authors are citing? Can the authors be more specific in their description of these “lines”. Maybe some representative images of a normal contralateral iris OCT would be helpful for comparison.  The authors seem to present the hyperreflective lines as a pivotal finding in their study, but the actual characterization and supportive figures are unclear.

The MIT value for the suspected melanosis cats was quite a bit lower than in the confirmed iris melanosis group. Why do the authors believe this occurred? Moreover, the MIT in the early FDIM group was almost identical to the confirmed melanosis group; and there were no statistically significant differences between groups.  This suggests that MIT is not a useful parameter.  The authors spend more time addressing PN ratio since that’s where they found a difference; but they do not mention the contradictory findings in the MIT part of the study.  This needs to be explored further or at the very least addressed by the authors.  

Were hyperreflective lines seen in any of the normal contralateral eyes? You didn’t include normal irises in your MIT analysis, but instead used the PN ratio. This would be important information to the reader.  Use of a normal control group for parameters other than PN ratio would help this manuscript.

Regarding Figure 4:  If “hyperreflective lines” are cited as being more characteristic of iris melanosis, what is the rationale?  Also, couldn’t angle of incidence of the OCT or differences in dilation status cause hyperreflective line artifacts?  Also, how are the authors correlating their imaging location with histologic section?  This is a very difficult process. Furthermore, the sectioning and fixation quality of the histologic image is poor in Panel A - the iris is fragmented and there is lack of normal anterior border layer which is likely artifactual and risks skewing evaluation of the areas of apparent proliferation.

Couldn’t dilation status of these eyes have been a significant confounding factor? The authors indicate that they used contralateral illumination; but how did these authors ensure that the intensity was the same for all cats?  Did they use a luminometer or other device? Also,  were all cats of the same temperament? Couldn’t sympathetic tone have also influenced pupil dilation status?

Discussion/Conclusions

The general information about OCT in the second paragraph of the discussion is largely unnecessary and distracting and would be better placed, and in more brief form, in the Introduction. Why all these details about swept source OCT when the authors used spectral domain?  This information is superfluous.

The authors cite that they evaluated the “entire iris” in eyes with melanosis and those with early FDIM; however, this is not true. They selected a portion of the iris to evaluate cross-sectionally. This terminology should be removed.

There is no peer-reviewed literature out there saying that mid-stage FDIM and advanced FDIM universally “manifest conspicuous iris thickening and irregularity” or that slit-lamp examination and ultrasound alone are sufficient. The authors provide no citation for this overstatement that is misleading to the reader.

In no way does this manuscript demonstrate the SD-OCT of the anterior segment is “adequate for evaluating the subtle structural differences between iris melanosis and early FDIM”. This is a vast overstatement of the value of the data from a very small, limited population, over half of which did not have confirmed diagnoses of melanosis.

Please provide references/citations for very broad and potentially inaccurate statements like, “SD-OCT has the significant advantage of faster image acquisition, which is crucial since animals often have difficulties maintaining eye fixation during OCT examination without anesthesia”.

In the Discussion, the authors indicate that their OCT data showed a 1.2 to 1.3-fold increase in iris thickness in early FDIM lesions compared to normal iris.  However, there is no mention in the Results of measured values for normal iris thickness. The only mention of  normal controls appears to be for determining PN ratio.  Are the authors referring to The owners cited in the Results a 1.2 to 1.3-fold increase for the PN ratio when comparing early FDIM lesions in comparison to the melanosis cats?  This is very confusing to the reader.

The authors cite that it is not practical to biopsy all areas of abnormal pigment which is true.  However, these authors described a methodology that ALSO does not include analysis of all areas and only “select” areas. Again, this methodology is highly susceptible to bias.

The comparison of photoreceptor receptor features on OCT has very little bearing on the characteristics of the anterior iris cells.

The authors cite reference 41; but no reference 41 is included in the references/bibliography.

The reasons in the Discussion for having a small sample size are not sufficient.  While some of the reasoning is valid, ultimately, a meaningful study requires more time and convenience sampling just to get something into the literature is not necessarily justifiable. The data presented here are not robust enough for clinical conclusions.

What “regional variations” are the authors referring to in the Discussion when talking about a single hospital study vs. a multi-institutional one?

The “exploratory nature” of this study is not a valid reason to not pursue sound standards in statistical analysis like appropriate use of non-parametric analyses or correction for multiple comparisons.

Formatting/Terminology Notes

Use of the termeyeball” as included in the Simple Summary is not the most appropriate terminology. Recommend use of “globe” or be even more accurate by saying “most common uveal tumor” or something to that effect.

Citing a textbook without providing the chapter and page numbers, as presented for references #1 and #2,  is not acceptable for a clinical/scientific paper.

Reviewer 2 Report

Comments and Suggestions for Authors

The study included in this manuscript describes the use of non-invasive contactless in-vivo imaging using OCT technology, to characterise possible changes within the iris of cats presenting pigmented iridal lesions. This is to try to present some features that could allow to differentiate benign melanosis and early signs of feline diffuse iris melanoma. 

This study is well conducted, well written and generally illustrated. It describes a novel use for OCT that is yet to be widely described. I believe this study may pave the way to better characterization of pigmented lesions in cats, and may facilitate the decision to pursue monitoring or be more aggressive with surgical intervention. 

Prior acceptation, I recommend:

1. add cross section and higher magnification image of the non affected iris if the eye was enucleated to better highlight the difference in presentation. This remains a little unclear to me base on your pictures in figure 4. Is there a chance that the quality of the magnified OCT image be improved?

2. The discussion part about SS-OCT vs SD-OCT can be removed. It is not relevant to this paper. 

3. Line 106-107 in the material and methods should be place within the result section

4. Please expand in your Material and Methods how the imaging was acquired (animal anesthetised or just held in front of the camera, any sedation used?)

5. Please consider adding a table to described which eye had histopathology performed, and if it was from a whole globe enucleated or a small biopsy, and what particular findings was seen by OCT

6. Is there any useful information following slit lamp evaluation that can be added? Focal thickenning of the tissue, were all the lesions flat on evaluation?

Reviewer 3 Report

Comments and Suggestions for Authors

Article synopsis: The goal of the study was to investigate the usefulness and feasibility of optical coherence tomography (OCT) in the staging of iris pigmented lesions in cats, particularly in differentiating between iris melanosis and feline diffuse iris melanoma (FDIM). Melanocytic neoplasms are the most common ocular neoplasms in cats, with FDIM being the most common form of feline ocular melanocytic neoplasia. FDIM is thus far only differentiated from benign melanosis via histopathology (requiring enucleation), and the clinical behavior of FDIM can be highly variable. Additional diagnostic measures may increase the ability for clinicians to identify the risk of malignancy and when enucleation is necessary. The performed OCT imaging in 18 cats with hyperpigmented iris lesions, of which 10 cats underwent enucleation for definitive diagnosis of the lesions within one month. Images were evaluated via ImageJ. The thickness of the iris in the pigmented lesion was contrasted to that of an anatomically comparable region in the iris of the contralateral eye. Although there was no significant difference in the thickness measurements of the lesions (most likely due to distribution of lesions across the varying thickness of the iris leaflets), early FDIM had a higher ratio of pigmented lesion to normal iris compared to the benign iris melanosis. A hyperreflective line was found in the benign lesions but not in the FDIM cases.

Review with comments: The study is scientifically sound and reproducible, and data appears to have been gathered using adequate ethical considerations. Statistical analyses are appropriate for the data. Citations within the paper are appropriate. Figures and graphs are of good quality and easy to understand.

The authors correctly identify several limitations, but their conclusions are appropriate for the scope of their study. A longitudinal study with more datapoints could allow for more confident assessments of the sensitivity and specificity of OCT in differentiating the benign versus malignant lesions, and I highly recommend the authors continue to pursue this in a future publication. As a pilot study, this paper serves as an interesting proof of concept for using OCT as a means of diagnosing early FDIM. Overall this paper is acceptable for publication with few corrections.

The introduction provides thorough background on iris melanosis, FDIM, and the challenges these diseases pose to clinicians and owners. I strongly recommend changing the statement in line 68 that enucleation risks a great decrease in the quality of life of a cat should they lose vision in the other eye (or need both eyes enucleated for some reason). As veterinarians it is important to convey to owners that their pet can adapt well to reduced vision or blindness and still have a long, happy life, particularly so when the consequences of delaying enucleation could mean metastatic disease and death.

In line 198, the claim should be “a significant relative increase in iris thickness” to more accurately convey that significance was found in the PN ratio rather than just the thickness of the iris.

In Figure 3, I recommend that titles be added above the graphs (i.e., MIT for 3A and PN for 3B) just to provide even more clarity.

Reviewer 4 Report

Comments and Suggestions for Authors

There are many limitation in this study.

First of all: there is a very low number of cases in each group.

This study is based on photos; the quality of those forme the OCT is very poor.

There is a photo (figure 2, photo D) in wich you gave a MiD-grade of melanosis but it's clear how that eye has a very high grade

Comments on the Quality of English Language

The quality of english is low

Round 2

Reviewer 1 Report

Comments and Suggestions for Authors

Dear Authors:

I have reviewed the submitted revised manuscript entitled, "A pilot study to evaluate the usefulness of optical coherence tomography for staging iris pigmented lesions in cats". I appreciate the revisions offered by the authors as well as the inclusion of reproducibility analysis. However, I still have concerns about the population selected to generate these data and therefore the conclusions drawn from those data. This reviewer urges these authors to remain patient and diligent in their recruitment efforts, particularly in the inclusion of eyes with confirmed histologic diagnoses. Please see my comments below.

The authors have provided some very useful, thorough, and appropriate clarifications to my many prior questions and I appreciate those efforts. My principal concerns, however, have not been resolved regarding the methods used. The AS-OCT imaging methods used are subject to bias. I do appreciate that the authors have acknowledged this. Frankly, those of us that use AS-OCT struggle with the same dilemma, namely how to develop a standardized method for anatomical mapping analysis of an ocular disease that is inherently irregular, heterogeneous, and frequently multifocal. It is not easy and some degree of bias may be very difficult to eliminate without a more robust method. Perhaps the authors can consider en face analysis of the iris lesions and geometric determination of the "lesion center" using a program like ImageJ. Again, there are many ways that this can be approached and none are perfect. Ultimately, despite the apparent reproducibility and precision cited by the authors, I still believe that the method presented for iris imaging and quantitative data collection is flawed and subject to inaccuracy and therefore is prone to misinterpretation. 

Above all else, I continue to have a major concern about the inclusion of cats with an unestablished diagnosis and a presumptive diagnosis of iris melanosis. This is the major flaw that I feel disqualifies any conclusions from being drawn. The predictive value of any diagnostic test requires verification of the endpoint; in this case, that endpoint is malignant transformation from iris melanosis to early FDIM. The crux of the clinical dilemma in feline ophthalmology and as identified by these authors in writing this manuscript is that differentiation of benign melanosis from malignant transformation and early FDIM is exceptionally difficult given the subtle nature of that change. Furthermore, the authors frame these data as of diagnostic value since they quantified a subtle but statistically-significant difference in iris thickness, but it is unclear and unsupported as to whether this is clinically significant. To begin to identify clinical significance would require histologic proof that the cases used to represent iris melanosis are indeed iris melanosis. If any of these cases happened to be histologically early FDIM, that would disqualify any conclusions drawn here.  

I am compelled by the identification of hyper-reflective lines provided by the authors - it is an interesting finding and may have some pertinence but as of now is a very "soft" finding. Again with such a small sample size, these descriptive data are very hard to interpret and would be confusing for a veterinary practitioner to understand without a more robust characterization.

Overall, I again applaud these authors for their efforts to bring some objective data to this frustrating dilemma. However, the conclusion that this method and these changes in iris thickness and the described hyperreflective lines are of immediate clinical utility to practitioners in the "staging" of feline pigmented iris lesions is not tenable. This reviewer understands that cross-sectional imaging modalities like AS-OCT have the potential to be diagnostically useful in the diagnosis and monitoring of this disease, and also that the authors are eager to share out their data and conclusions. However, this data cannot "move the needle" for this disease without more case numbers and specifically, inclusion of cases that have lesions with confirmed histopathology. This reviewer understands that recruitment for such studies is lengthy and often complicated and may take many years; but this process is required of all clinician scientists to generate data that is sound and can be more clearly interpreted. This study is not yet powered sufficiently to achieve the stated goal and this reviewer urges the authors to extend their efforts to include a larger and more suitable population. 

Reviewer 2 Report

Comments and Suggestions for Authors

I do not see any of the comments that I have made addressed by the authors.

Author Response

We sincerely apologize for any confusion caused by our response to the revision instructions. Upon reviewing our previous correspondence, we realize that we inadvertently addressed a different reviewer’s questions instead of responding to the specific comments you provided. We have responded to the your first comments in a point-by-point manner, and diligently addressed each comment and revised our manuscript accordingly.

Reviewer 4 Report

Comments and Suggestions for Authors

Regarding the quality of the pictures we understand your difficulty in obtaining a good OCT immages in not sedated animals. You should include this point in the limitations.

Comments on the Quality of English Language

the english is improved.
